# Non-Alcoholic Fatty Liver Disease Is Associated with Higher Metabolic Expenditure in Overweight and Obese Subjects: A Case-Control Study

**DOI:** 10.3390/nu11081830

**Published:** 2019-08-07

**Authors:** Rosa Reddavide, Anna Maria Cisternino, Rosa Inguaggiato, Ornella Rotolo, Iris Zinzi, Nicola Veronese, Vito Guerra, Fabio Fucilli, Giuseppe Di Giovanni, Gioacchino Leandro, Sara Giannico, Maria Gabriella Caruso

**Affiliations:** 1Ambulatory of Clinical Nutrition, National Institute of Gastroenterology, Research Hospital, IRCCS “Saverio de Bellis” of Castellana Grotte (BA), 70013 Castellana Grotte, Italy; 2Unit of Epidemiology and Biostatistical, National Institute of Gastroenterology, Research Hospital, IRCCS “Saverio de Bellis” of Castellana Grotte (BA), 70013 Castellana Grotte, Italy; 3Unit of Radiology, National Institute of Gastroenterology, Research Hospital, IRCCS “Saverio de Bellis” of Castellana Grotte (BA), 70013 Castellana Grotte, Italy; 4Unit of Gastrenterology 1, National Institute of Gastroenterology, Research Hospital, IRCCS “Saverio de Bellis” of Castellana Grotte (BA), 70013 Castellana Grotte, Italy

**Keywords:** NAFLD, metabolism, obesity

## Abstract

Non-alcoholic fatty liver disease (NAFLD) is a common condition in Western countries. However, their metabolic characteristics are poorly known even though they could be important. Therefore, the objective of this study was to measure resting metabolic parameters in overweight/obese adults with hepatic steatosis compared to controls, matched for age, sex, and obesity level. Hepatic steatosis was diagnosed with liver ultrasound. Energy metabolism was measured with indirect calorimetry: energy expenditure (REE), predicted REE, the ratio between REE and the predicted REE, and the respiratory quotient (RQ) were reported. We measured some anthropometric, body composition, and bio-humoral parameters; 301 participants with NAFLD were matched for age, sex, and obesity level with 301 participants without NAFLD. People with NAFLD showed significantly higher REE (1523 ± 238 vs. 1464 ± 212 kcal, *p* = 0.005), REE/REE predicted ratio (98.2 ± 9.4 vs. 95.7 ± 8.1, *p* = 0.002), and RQ (0.88 ± 0.08 vs. 0.85 ± 0.07, *p* = 0.03). Moreover, the NAFLD group had significantly higher inflammatory and insulin-resistance parameters compared to controls. In conclusion, NAFLD is associated with a significantly higher metabolic expenditure, as measured with indirect calorimetry, compared to a similar cohort of individuals without this condition. Higher inflammatory levels in patients with NAFLD can probably explain our findings, even if other research is needed on this issue.

## 1. Introduction

Non-alcoholic fatty liver disease (NAFLD) is probably the most common chronic liver disease, particularly in Western industrialized countries, in which some data suggest a prevalence of at least 20% in adults [1]. NAFLD is traditionally associated with a metabolic syndrome, and consequently insulin resistance, and often involves abnormal glucose and cholesterol metabolism parameters [2,3].

Since NAFLD is often associated with obesity and metabolic parameters, some guidelines suggest the assessment of body composition and metabolic parameters in all people affected by this condition [4]. Several risk factors are traditionally known to affect NAFLD including a Western diet, high fat and cholesterol diet, high fructose diet, oxidative stress, and alcohol, even within the normal range of intake [5,6,7,8,9,10].

Indirect calorimetry is a common method used in physiological testing and clinical practice since it enables easy and non-invasive evaluation of energy metabolism in real time [11]. However, the studies that have so far explored the metabolic parameters in people with NAFLD, as assessed by indirect calorimetry are, to the best of our knowledge, very much limited by small sample sizes and by the fact that controls were not matched for some important confounders in indirect calorimetry, such as obesity level [12]. In an exploratory analysis, for example, the authors found that in 20 patients with NAFLD, the observed significantly lower metabolic expenditure was mainly due to lower fat-oxidation in basal conditions [13]. Even if this finding is important, the study did not match for any confounders and the sample size was limited [13]. However, a better knowledge of the metabolic characteristics of people affected by NAFLD is useful since calorimetric parameters help to indicate the best approach to decreasing obesity levels in these subjects. Given this background, the objective of this study was to measure metabolic parameters during rest, employing indirect calorimetry in adult patients with NAFLD compared to controls, matched for several factors, including obesity level.

## 2. Materials and Methods

### 2.1. Participants

This study included women and men who successively accessed our Outpatients Obesity Center in the Istituto di Ricovero e Cura a Carattere Scientifico (IRCCS, Institute of Recovery and Cure with Scientific Characterization) of Castellana Grotte, a town in Southern Italy (Apulia region).

We successively enrolled overweight/obese subjects, as defined by World Health Organization criteria [14], aged at least 18 years, non-smokers, and without major comorbidities based on self-reported information or from their medical history (e.g., severe renal failure or hepatic failure, cancer in the previous 5 years, diabetes treated with insulin), between March 2015 and March 2018. People with active chronic hepatitis (e.g., HBV or HCV), autoimmune hepatitis, or hemochromatosis (based on self-reported information), as well as people with an elevated alcohol intake, were excluded from this study.

Overall, in the period between March 2015 and March 2018, 3336 patients were evaluated in our Center. Applying the above-listed inclusion and exclusion criteria, 301 patients with NAFLD were compared to 301 controls (i.e., not affected by NAFLD) based on age, gender, and body mass index (BMI), with a tolerance of 3 years and of 1 kg/m^2^ for age and BMI, respectively.

The proposal of this research was approved by the Institutional Review Board (Ethical Committee) of IRCCS De Bellis and written informed consent, given during the visit, was obtained from each participant before enrolment in the study.

### 2.2. NAFLD Diagnosis

All the participants underwent a standardized ultrasound examination made by two trained investigators using a Hitachi H21 Vision (Hitachi Medical Corporation, Tokyo, Japan). Examination of the visible liver parenchyma was performed with a 3.5 MHz transducer. A scoring system was adopted to obtain a semi-quantitative evaluation of fat in the liver [15,16,17,18].

The degree of liver fatty infiltration was then graded according to the appearance of the liver echotexture, the liver echo penetration, and the clarity of the liver blood vessels, as well as the liver diaphragm differentiation in the echo amplitude. A score was assigned to each criterion, indicating the level of fatty liver accumulation. A score of 2 indicated definite positive (++) fatty liver infiltration, a score of 1 indicated probably positive (+) fatty liver infiltration, and a score of 0 indicated the absence of fatty liver (−). The fatty liver score ranged from 0 to 6, higher values indicating a greater severity. The intra- and interobserver variability of the partial and total scores obtained for the assessment of hepatic fat by two trained radiologists was observed in 32 consecutive subjects, and ranged from 0.77 to 0.85 [15]. Appendix A reports further criteria used for this methodology.

NAFLD was then diagnosed as the presence of liver steatosis together with alcohol consumption <30 g/d for men and <20 g/d for women [19]. Alcohol intake was determined using the food-frequency questionnaire based on the previous 12 months, using the intake of wine, beer, and super-alcoholic drinks as parameters.

### 2.3. Indirect Calorimetry

Energy metabolism was measured by indirect calorimetry (Aero Monitor AE-300s; Minato Medical Science, Osaka, Japan). We used the method reported in Simonson and DeFronzo [20] to measure oxygen uptake and carbon dioxide exhalation under resting and fasting conditions in the early morning.

The resulting values were then used to calculate resting energy expenditure (REE), predicted REE, and the ratio between REE and the predicted REE. The respiratory quotient (RQ) was also reported [21].

### 2.4. Other Measurements

In our Center, we routinely collected several anthropometric and bio-humoral parameters. Body weight was measured to the nearest 0.1 kg using a standard scale with subjects wearing light clothing and no shoes, barefoot standing height was measured to the nearest 0.1 cm using a wall-mounted stadiometer, and body mass index (BMI) was calculated as weight in kilograms divided by height in meters squared. Body composition was assessed with the bioimpendance analysis (BIA 101 Anniversary AKERN/RJL Systems) emitting an alternating sinusoidal electric current of 400 μA at an operating single frequency of 50 kHz. With these parameters, we calculated fat mass and fat-free mass, reported in kg.

We routinely collected data on several bio-humoral parameters, namely complete blood count, albumin, renal function, serum uric acid, liver enzymes, cholesterol parameters, insulin, fasting plasma glucose, and thyroid stimulating hormone (TSH) with standardized procedures.

### 2.5. Statistical Analysis

Data are reported as means ± standard deviations (SDs) for quantitative measures and as percentages for all categorical variables. The *p*-values were calculated using the independent *t*-test for continuous variables and the chi-square test for categorical ones. In each group, normal distributions of continuous variables were tested using the Kolmogorov–Smirnov test.

A linear regression model, adjusted for age and sex, reporting adjusted mean values of biomarkers concentrations were calculated across the six categories of the severity score, with a *p*-value for the linear trend.

All analyses were performed using SPSS 17.0 for Windows (SPSS Inc., Chicago, IL, USA). All statistical tests were two-tailed and significance was set at a *p*-value < 0.05.

## 3. Results

The anthropometric parameters are fully reported in Table 1. As expected, they showed no significant differences in fat-free mass (*p* = 0.67) and fat mass (*p* = 0.30).

The NAFLD group, however, showed significantly higher values compared to controls for white blood cells, hemoglobin, albumin, serum uric acid, and hepatic enzymes (all values *p* < 0.05) (Table 2). Similarly, people with NAFLD had worse metabolic control (as shown by worse values for cholesterol, triglycerides, and glucose metabolism parameters), and significantly higher values for TSH.

Compared to controls, people with NAFLD showed significantly higher REE (1523 ± 238 vs. 1464 ± 212 kcal, *p* = 0.005), REE/REE predicted ratio (98.2 ± 9.4 vs. 95.7 ± 8.1, *p* = 0.002), and respiratory quotient (0.88 ± 0.08 vs. 0.85 ± 0.07, *p* = 0.03), as reported in Table 3.

Table 4 shows the association between the severity of NAFLD and single biomarker concentrations using a linear regression model and after adjusting for age and sex. This analysis confirmed that people with a higher severity of NAFLD had significantly higher values of WBC, hemoglobin, creatinine, serum uric acid, and hepatic enzymes, as well as worse metabolic and cholesterol profiles.

## 4. Discussion

This study including 301 patients with NAFLD compared to 301 controls matched for age, sex, and BMI (including total fat mass), showed that people with NAFLD had significantly higher metabolic expenditure as measured with indirect calorimetry. To the best of our knowledge, our study is one of the first to report metabolic features in people with NAFLD.

In our study, people with NAFLD, even if matched for obesity status, showed significantly higher inflammatory levels, as shown by higher white cells values. Overall, we can hypothesize that intra-hepatic accumulation of fat was able to increase peripheral inflammatory levels. Hepatocyte apoptosis probably played a pivotal role in this disease and, likely, was the first step in promoting inflammation in these individuals [22]. However, this remains a speculation since we do not have data regarding inflammatory parameters, except for white blood cells. In this regard, we found that serum albumin levels were significantly higher in NAFLD patients compared to controls, in accordance with the recent literature on this topic [23]. In this regard, free-fatty-acid-induced toxicity probably represents one of the mechanisms in the pathogenesis of NAFLD, promoting the passage from NAFLD to non-alcoholic steato-hepatitis (NASH) [22,24]. Inflammation is significantly related to a higher energy expenditure [25]. This finding can be justified using several explanations. First, inflammation is associated with an elevated VO2 [26], enhanced lipolysis and fat utilization [27], high concentration of catabolic hormones, and extensive protein catabolism [28]. Moreover, maintaining the immune function accounts for about 15% of daily energy expenditure [29].

Another interesting finding in our work is that a greater severity of NAFLD was significantly associated with higher REE values, indicating that people with more severe forms of NAFLD had a higher energy expenditure. This finding suggests that higher steatosis levels were significantly associated with a greater metabolic expenditure. Unfortunately, this finding is weak, as shown by the correlation coefficient, and other studies are needed to confirm our results.

We believe that our findings may have an important bearing on the clinical management of obese subjects and those with NAFLD. Clearly, people with a greater REE should lose weight faster than those with lower values. However, it should be noted that the metabolic adaptation of our body to weight loss interventions mainly depends on the type of intervention proposed. Calorie restriction, for example, can cause significant decreases in all the energy expenditure components [30], mainly due to a decrease in fat-free mass. On the contrary, physical exercise is able to increase the metabolic parameters [31]. At the same time, we should consider the important role of adaptive thermogenesis, which refers to changes in REE and non-resting energy expenditure that are independent of changes in body composition [32]. Therefore, when we approach patients with NAFLD, we should note that, even if they have a higher REE, they also have a higher RQ, indicating that these people probably consume more proteins than carbohydrates compared to their lower REE counterparts [33]. Overall, our experience with these patients, based on the calorimetric findings, suggests that the best approach toward achieving weight loss in these people could be a low-calorie diet with a limited amount of proteins. However, specific studies in body composition changes after weight loss in NAFLD patients are needed to better understand the best solution for these patients.

The strengths of the present study include the large sample size and the fact that we compared cases and controls on the basis of several parameters, in particular obesity. However, the findings described herein should be interpreted within some limitations. First, we have data only on resting parameters and not after exercise because we collected these data for clinical purposes only. Second, the diagnosis of NAFLD was made only using echography, whilst the gold standard for this diagnosis is a liver biopsy [34]. Third, food intake has not been taken into account, but this information could add important information regarding energy intake and expenditure. Fourth, body composition was analyzed through BIA, but other tools (such as Dual-energy X-ray absorptiometry or magnetic resonance or computerized tomography) are more precise in estimating muscle and fat mass. Furthermore, hepatic steatosis was estimated with ultrasounds, whilst, again, MRI with PDFF (proton density fat fraction) and MRS (magnetic resonance spectroscopy) can give better estimates of hepatic fat being less observatory dependent [35]. Finally, we have data only on white blood cells, whilst other inflammatory markers (e.g., serum C reactive protein) were not measured, nor did we assess the oxidative/glycolytic fluxes in isolated PBMCs (peripheral blood mononuclear cells), that could further complete and support the relevance of our findings.

## 5. Conclusions

Our data suggest that NAFLD is associated with a significantly higher metabolic expenditure, as measured by indirect calorimetry, compared to a similar cohort of individuals without NAFLD. Higher inflammatory levels in patients with NAFLD can probably explain our findings, even if other research is needed on this issue. Finally, changes in values of metabolic parameters after weight loss intervention are also of great interest and should be explored via specific studies including only people with NAFLD.

## Figures and Tables

**Table 1 nutrients-11-01830-t001:** Anthropometric parameters related to the presence or not of hepatic steatosis.

Parameter	NAFLD (*n* = 301)	Controls (*n* = 301)	*p*-Value
Age (years)	51.9 (12.2)	51.5 (12.5)	0.72
Female gender (n, %)	246 (81.7)	246 (81.7)	1.00
Weight (kg)	83.8 (12.0)	83.1 (12.0)	0.52
BMI (kg/m^2^)	32.3 (4.0)	32.1 (3.9)	0.59
Fat-free mass (kg)	53.4 (8.9)	53.1 (8.9)	0.67
Fat mass (kg)	30.7 (7.8)	30.0 (7.9)	0.30

**Table 2 nutrients-11-01830-t002:** Bio- humoral parameters related to the presence or not of hepatic steatosis.

Parameter	NAFLD (*n* = 301)	Controls (*n* = 301)	*p*-Value
White cells (n/10^6^ L)	6745 (2031)	5964 (2376)	<0.0001
Hemoglobin (g/dL)	13.74 (1.91)	12.85 (3.38)	<0.0001
Albumin (g/L)	3.19 (1.74)	1.99 (2.12)	0.001
Creatinine (μg/L)	0.79 (0.20)	0.74 (0.29)	0.02
Serum uric acid (mg/dL)	4.57 (1.28)	4.07 (1.56)	<0.0001
GPT	31.8 (22.3)	24.7 (20.2)	<0.0001
GGT	27.2 (20.3)	22.7 (26.0)	0.03
Total cholesterol (mg/dL)	200 (41)	189 (59)	0.01
HDL (mg/dL)	54 (15)	55 (20)	0.53
Triglycerides (mg/dL)	129 (76)	94 (49)	<0.0001
Fasting plasma glucose (mg/dL)	102 (28)	89 (28)	<0.0001
Insulin	13.7 (15.5)	9.3 (7.9)	<0.0001
TSH	2.04 (1.36)	1.82 (1.29)	0.05

**Abbreviations:** GPT: glutamic pyruvic transaminase; GGT: Gamma-glutamyl transferase; TSH: thyroid stimulating hormone.

**Table 3 nutrients-11-01830-t003:** Calorimetric parameters related to the presence or not of hepatic steatosis.

Parameter	NAFLD (n = 301)	Controls (n = 301)	*p*-Value
REE (kcal)	1523 (238)	1464 (212)	0.005
Predicted REE (kcal)	1552 (207)	1531 (204)	0.28
REE/REE predicted	98.2 (9.4)	95.7 (8.1)	0.002
Respiratory quotient	0.88 (0.08)	0.85 (0.07)	0.03

**Abbreviations:** REE: resting energy expenditure.

**Table 4 nutrients-11-01830-t004:** Linear trend of single biomarker concentrations on linear regression model of the severity score, adjusted for age and gender.

		NAFLD Score (%)							
Parameter (Mean ± Standard Error)	Sample Size	0 (80.73%)	1 (0.26%)	2 (0.78%)	3 (5.21%)	4 (4.69%)	5 (2.60%)	6 (5.73%)	*p*-Value for Trend
White cells (n/10^6^ L)	356	6670.42 ± 110.33	6480.77 ± 1875.31	6752.17 ± 1083.65	7163.04 ± 485.90	6149.92 ± 472.72	6610.26 ± 592.73	7199.73 ± 415.29	<0.001
Hemoglobin (g/dL)	356	13.57 ± 0.07	13.25 ± 1.22	14.19 ± 0.70	13.97 ± 0.31	13.85 ± 0.31	14.09 ± 0.38	14.32 ± 0.27	<0.001
Albumin (g/L)	353	4.16 ± 0.14	3.95 ± 0.12	3.73 ± 0.12	3.52 ± 0.15	3.31 ± 0.19	3.09 ± 0.23	2.88 ± 0.28	<0.001
Creatinine (μg/L)	353	0.78 ± 0.01	0.75 ± 0.16	0.94 ± 0.09	0.86 ± 0.04	0.70 ± 0.04	0.76 ± 0.05	0.75 ± 0.03	<0.001
Serum uric acid (mg/dL)	329	4.38 ± 0.07	4.29 ± 1.11	5.30 ± 0.64	4.60 ± 0.30	4.90 ± 0.29	5.33 ± 0.37	4.65 ± 0.26	<0.001
GPT (U/L)	351	28.78 ± 1.34	27.08 ± 22.57	28.43 ± 13.04	27.03 ± 5.68	36.00 ± 5.69	46.13 ± 7.13	39.12 ± 5.00	<0.001
GGT (U/L)	340	25.15 ± 1.32	18.51 ± 21.79	26.94 ± 12.59	29.20 ± 5.48	35.13 ± 5.49	31.65 ± 7.26	36.55 ± 4.83	<0.001
Total cholesterol (mg/dL)	350	200.82 ± 2.26	239.82 ± 38.03	195.66 ± 21.97	198.30 ± 9.57	190.83 ± 9.59	211.85 ± 12.02	200.24 ± 8.63	<0.001
HDL (mg/dL)	343	56.93 ± 0.78	71.25 ± 12.99	44.44 ± 7.50	51.49 ± 3.27	47.99 ± 3.27	50.27 ± 4.10	48.25 ± 2.95	<0.001
Triglycerides (mg/dL)	350	114.85 ± 3.93	93.86 ± 66.09	145.19 ± 38.19	134.57 ± 16.63	134.84 ± 16.66	186.33 ± 20.89	152.40 ± 15.00	<0.001
Fasting plasma glucose (mg/dL)	354	100.55 ± 1.54	96.50 ± 26.04	97.09 ± 15.05	93.09 ± 6.55	106.45 ± 6.56	102.26 ± 8.23	94.03 ± 5.89	<0.001
Insulin (mIU/L)	323	12.70 ± 0.90	6.00 ± 14.59	16.81 ± 8.43	13.44 ± 3.78	12.46 ± 3.68	19.20 ± 4.86	13.56 ± 3.47	<0.001
TSH (µIU/mL)	342	1.98 ± 0.08	1.61 ± 1.32	1.74 ± 0.76	2.56 ± 0.33	2.39 ± 0.33	1.79 ± 0.44	1.44 ± 0.31	<0.001

**Abbreviations:** GPT: glutamic pyruvic transaminase; GGT: Gamma-glutamyl transferase; NAFLD: non-alcoholic fatty liver disease; TSH: thyroid stimulating hormone.

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
