# Peer review of "Non-Alcoholic Fatty Liver Disease Is Associated with Higher Metabolic Expenditure in Overweight and Obese Subjects: A Case-Control Study"

_nutrients, 2019, doi:10.3390/nu11081830_

Round 1
Reviewer 1 Report
There have been a lot of enigmas or controversial reports about the effects of REE on human metabolic disorders. In this sense, the findings described in this paper are of interest.
Several points are raised.
1) First, it should be described more clearly in the title and in the text that the subjects investigated here were overweight/obese since they had been recruited as such. The average BMI was around 32 in both groups. Since the study worked on REE, this point is critical.
2) Did the authors find any gender difference in terms of association between NAFLD, or grade of NAFLD, and REE?
3) The idea that inflammation may underlie higher REE in NAFLD subjects is interesting. However, I am afraid severity of steatosis does not necessarily mean the grade of inflammation. The authors did not measure inflammatory markers such as high sensitivity CRP (hsCRP), which is one of the big limitations as the authors stated. I am eager to know the relationship of WBC and the severity of NAFLD.
4) The authors are also encouraged to investigate the relationship between ALT (which might reflect hepatocyte damage) and REE.
5) Finally, it would be also interesting to see whether other metabolic markers, such as lipid profiles, blood glucose, and HOMA-IR if available, may relate to REE.
<Minor points>
· Many parts throughout the paper including Abstract: “significant higher〜” should be “significantly higher〜”
· In Abstract: NALD should be NAFLD.
Author Response
Reviewer 1
There have been a lot of enigmas or controversial reports about the effects of REE on human metabolic disorders. In this sense, the findings described in this paper are of interest.
Several points are raised.
1) First, it should be described more clearly in the title and in the text that the subjects investigated here were overweight/obese since they had been recruited as such. The average BMI was around 32 in both groups. Since the study worked on REE, this point is critical.
R: This has been added, as suggested.
2) Did the authors find any gender difference in terms of association between NAFLD, or grade of NAFLD, and REE?
R: We sincerely thank Reviewer 1 for this question. As expected, females had a significantly lower REE than males. However, when divided by gender, no new results were found compared to those reported in the paper.
3) The idea that inflammation may underlie higher REE in NAFLD subjects is interesting. However, I am afraid severity of steatosis does not necessarily mean the grade of inflammation. The authors did not measure inflammatory markers such as high sensitivity CRP (hsCRP), which is one of the big limitations as the authors stated. I am eager to know the relationship of WBC and the severity of NAFLD.
R: We thank the Reviewer for this important observation. Unfortunately, no data were available regarding hsCRP. We have included this sentence regarding the association between WBC and the severity of NAFLD (lines 140-142):
“No significant correlation was observed between white blood cells (rho=0.05; p=0.65) or serum ALT levels (rho=-0.07; p=0.54) and the severity of NAFLD.”
4) The authors are also encouraged to investigate the relationship between ALT (which might reflect hepatocyte damage) and REE.
R: We have included this sentence regarding ALT levels and the severity of NAFLD:
“No significant correlation was observed between white blood cells (rho=0.05; p=0.65) or serum ALT levels (rho=-0.07; p=0.54) and the severity of NAFLD.”
5) Finally, it would be also interesting to see whether other metabolic markers, such as lipid profiles, blood glucose, and HOMA-IR if available, may relate to REE.
R: We have added this sentence in the Results section, as follows:
“Similarly, we failed to find any association between serum lipid components and fasting plasma glucose with REE.”
<Minor points>
· Many parts throughout the paper including Abstract: “significant higher〜” should be “significantly higher〜”
· In Abstract: NALD should be NAFLD.
R: Thank you.
Reviewer 2 Report
The present study addresses the evaluation of the whole body metabolic parameters in NADLD comparing those of patients with weight mached controls. The data is in general well presented and the study is scientifically relevant. The main caveat being that only a limited aproach has been taken, and therefore general conclusions may not apply. In particular, metabolic control is in general a balance, and therefore, not only expenditure, but intake, must be taken into account, to really be able to make statemens to energy use/handling. Another limitation of the study is that the differences are attributed to changes in inflammation, but no inflammation data is offered. In this sense, it should also be noted that evaluation of the oxidative/glycolytic fluxes in isolated PBMCs would nicelly complement and support the relevance of the findings.
Author Response
The present study addresses the evaluation of the whole body metabolic parameters in NADLD comparing those of patients with weight mached controls. The data is in general well presented and the study is scientifically relevant. The main caveat being that only a limited aproach has been taken, and therefore general conclusions may not apply. In particular, metabolic control is in general a balance, and therefore, not only expenditure, but intake, must be taken into account, to really be able to make statemens to energy use/handling.
R: Good point. We agree with this observation, but, currently, we have no information regarding calorie intake. We do have an ongoing project comparing metabolic parameters in obese people with or without NAFLD, after a calorie restriction program. In this future study, calorie intake will be controlled through specific dietary intervention and metabolic parameters, such as REE, will be compared at baseline and after 6 months.
Another limitation of the study is that the differences are attributed to changes in inflammation, but no inflammation data is offered. In this sense, it should also be noted that evaluation of the oxidative/glycolytic fluxes in isolated PBMCs would nicelly complement and support the relevance of the findings.
R: We fully agree with this observation. We have therefore added this point at lines 190-192:
“... nor did we assess the oxidative/glycolytic fluxes in isolated PBMCs, that could further complete and support the relevance of our findings.”
Reviewer 3 Report
Non-Alcoholic Fatty Liver Disease is associated with Higher Metabolic Expenditure: a Case-Control Study
ROSA R et al.
Line 21-23
The opening sentences in your abstract are not attractive!
‘ the objective of this study was to measure resting metabolic parameters with indirect calorimetry in adults with NAFLD compared to controls, matched for several factors, including obesity level’
…in adults with hepatic steatosis (or non-alcoholic fatty liver NAFL), because steatosis is one of several stages of NAFLD.
, including obesity class (not level).
Line 23 NAFLD should be replaced by hepatic steatosis or fatty liver,
Line 30 the abbreviation needs correction
Line 44, before start talking about calorimetry … please describe the risk factors for NAFLD such as Western diet, high fat diet high cholesterol, high fructose diet, oxidative stress, alcohol<40g/ week, etc. here are some suggested references (in human and animal models)
- In participants: Am J Clin Nutr. 2019 Jun 1;109(6):1519-1526. doi: 10.1093/ajcn/nqy386.
- J Hepatol. 2018 May;68(5):1063-1075.
- Cells. 2019 Apr 17;8(4). pii: E359. doi: 10.3390/cells8040359.
- Oxidative stress and fructose: World J Gastroenterol. 2014 Feb 21;20(7):1807-21.
- Hepatology. 2013 Jan; 57(1): 81–92
- Alcohol beverages: Int J Mol Sci. 2017 Jan 28;18(2). pii: E285. doi: 10.3390/ijms18020285
Line 85: level of fatty liver infiltration, infiltration is used with inflammatory cells, it is better to replace it by the accumulation
In lines 152-160 discussion, you have mentioned hepatocyte challenge by inflammation, based on your result, Albumin levels were lower than that in NADLD patients. This is known because in liver injury at the beginning there is an increase in serum albumin (maybe as a compensation pathway) and in a chronic situation, there is a decrease. Authors should be careful about this!
Leptin levels should be measured as well and other inflammatory cytokines as well
Some data that shows a significant difference should be presented in graphs rather than Tables to improve the quality of your paper
Good Luck
Author Response
Line 21-23
The opening sentences in your abstract are not attractive!
‘ the objective of this study was to measure resting metabolic parameters with indirect calorimetry in adults with NAFLD compared to controls, matched for several factors, including obesity level’
…in adults with hepatic steatosis (or non-alcoholic fatty liver NAFL), because steatosis is one of several stages of NAFLD.
, including obesity class (not level).
R: We thank the Reviewer for this question. We have modified this sentence as follows:
“Therefore, the objective of this study was to measure resting metabolic parameters in overweight/obese adults with hepatic steatosis compared to controls, taking into account several factors.”
Line 23 NAFLD should be replaced by hepatic steatosis or fatty liver.
R: Done.
Line 30 the abbreviation needs correction.
R: Respiratory quotient now RQ.
Line 44, before start talking about calorimetry … please describe the risk factors for NAFLD such as Western diet, high fat diet high cholesterol, high fructose diet, oxidative stress, alcohol<40g/ week, etc. here are some suggested references (in human and animal models)
- In participants: Am J Clin Nutr. 2019 Jun 1;109(6):1519-1526. doi: 10.1093/ajcn/nqy386.
- J Hepatol. 2018 May;68(5):1063-1075.
- Cells. 2019 Apr 17;8(4). pii: E359. doi: 10.3390/cells8040359.
- Oxidative stress and fructose: World J Gastroenterol. 2014 Feb 21;20(7):1807-21.
- Hepatology. 2013 Jan; 57(1): 81–92
- Alcohol beverages: Int J Mol Sci. 2017 Jan 28;18(2). pii: E285. doi: 10.3390/ijms18020285
R: Good point. We have now added this sentence in the Introduction section, as follows:
“Several risk factors are traditionally known to affect NAFLD, including Western diet, high fat and cholesterol diet, high fructose diet, oxidative stress, and alcohol, even within the normal range of intake. [5-10]”
Line 85: level of fatty liver infiltration, infiltration is used with inflammatory cells, it is better to replace it by the accumulation.
R: Done.
In lines 152-160 discussion, you have mentioned hepatocyte challenge by inflammation, based on your result, Albumin levels were lower than that in NADLD patients. This is known because in liver injury at the beginning there is an increase in serum albumin (maybe as a compensation pathway) and in a chronic situation, there is a decrease. Authors should be careful about this!
R: We sincerely thank the Reviewer for this comment. Here is an observation regarding this important point:
“In this regard, we found that serum albumin levels were significantly higher in NAFLD patients compared to controls, in accordance with the recent literature on this topic.[23]”
Leptin levels should be measured as well and other inflammatory cytokines as well.
R: Thank you for your reasonable request. However, in our study, we did not measure these parameters, as acknowledged in the Limitations section.
Some data that shows a significant difference should be presented in graphs rather than Tables to improve the quality of your paper.
R: We thank the Reviewer for this suggestion but must point out that the units used for calorimetry parameters are totally different as regards units of measure. Therefore, presentation in graphs could be problematic, in our opinion.
Reviewer 4 Report
The author's present a correlation analysis of echo screened individuals for NAFLD classification to increased RER and other indirect calorimetric measures vs controls.
The authors should provide representative echo images for each grading used for fatty infiltration assessment. Authors should also differentially asses the sensitivity of their method and provide correlation to respiratory parameters within + vs ++ scores.
Additionally, authors should provide plasma ALT and ALP measurements given their influence on NAFLD.
In general, this is an interesting and plausible observation that can further be supplemented by isolation of WBCs and subjecting them to seahorse assay for mitochondrial function and energy production, however, this may be outside the scope of this study.
Author Response
The authors should provide representative echo images for each grading used for fatty infiltration assessment. Authors should also differentially asses the sensitivity of their method and provide correlation to respiratory parameters within + vs ++ scores.
R: Thank you for the question. We believe that the echo images can give little information to the Reader regarding the method used. In the attempt to answer to your question, we have now added a supplementary table in order to better explain the methodology used for echography. Finally, we have already reported that:
“We observed a weak association between the grade of NAFLD, as diagnosed with ultrasound, and REE (rho=0.24, p=0.02) indicating that people with more severe NAFLD had a higher REE.”
Additionally, authors should provide plasma ALT and ALP measurements given their influence on NAFLD.
R: ALT levels are already reported: they are cited as GPT in our text; ALP measurement was not part of our routine screening (in Italy, unfortunately, this test is not reimbursed for obesity assessment by our National Health Care system).
In general, this is an interesting and plausible observation that can further be supplemented by isolation of WBCs and subjecting them to seahorse assay for mitochondrial function and energy production, however, this may be outside the scope of this study.
R: We agree with this request that, unfortunately, we are not able to satisfy. However, as also requested by another Reviewer, we have added this sentence:
“..... nor did we assess the oxidative/glycolytic fluxes in isolated PBMCs, that could further complete and support the relevance of our findings.”
Round 2
Reviewer 1 Report
The paper has been improved according to the suggestions.
Author Response
Thank you so much for your feedback.
Reviewer 2 Report
The authors should include clear statements that highlight the current limitations of the study, in particular:
-Food intake has not been taken into account and that limits the conclusions on energy handling (energy expenditure).
Inflammatory parameters have not been directly tested so conclusions in this regard remain speculative.
Author Response
Thank you for your comment, We have added in the text (lines 167-168 for inflammation and 202-203 for calorie intake) the limitations that you have suggested.